# The Effect of Silver Nanoparticles on the Digestive System, Gonad Morphology, and Physiology of Butterfly Splitfin (*Ameca splendens*)

**DOI:** 10.3390/ijms241914598

**Published:** 2023-09-27

**Authors:** Maciej Kamaszewski, Kacper Kawalski, Wiktoria Wiechetek, Hubert Szudrowicz, Jakub Martynow, Dobrochna Adamek-Urbańska, Bogumił Łosiewicz, Adrian Szczepański, Patryk Bujarski, Justyna Frankowska-Łukawska, Aleksander Chwaściński, Ercüment Aksakal

**Affiliations:** 1Institute of Animal Sciences, Warsaw University of Life Sciences, Warsaw, Ciszewskiego 8, 02-786 Warsaw, Poland; s203060@sggw.edu.pl (K.K.); wiktoria_wiechetek@sggw.edu.pl (W.W.); s203069@sggw.edu.pl (J.M.); dobrochna_adamek@sggw.edu.pl (D.A.-U.); s213028@sggw.edu.pl (B.Ł.); adrian_szczepanski@sggw.edu.pl (A.S.); s203046@sggw.edu.pl (P.B.); s213006@sggw.edu.pl (A.C.); 2Institute of Biology, Warsaw University of Life Sciences, Warsaw, Nowoursynowska 159, 02-776 Warsaw, Poland; justyna_frankowska-lukawska@sggw.edu.pl; 3Division of Animal Biotechnology, Department of Agricultural Biotechnology, Agriculture Faculty, Akdeniz University, Antalya 07058, Turkey; aksakal@akdeniz.edu.tr

**Keywords:** enzymatic activity, Goodeidae fishes, histology, nanoecotoxicology, silver nanoparticles

## Abstract

The aim of this study was to determine the effects of silver nanoparticles (AgNPs) on the morphology and enzymatic activity of butterfly splitfin (*Ameca splendens*). Individuals of both sexes, aged about five months, were exposed to AgNPs at concentrations of 0 (control group), 0.01, 0.1, and 1.0 mg/dm^3^ for 42 days. On the last day of the experiment, the fish were euthanized, subjected to standard histological processing (anterior intestine, liver, and gonads), and analysed for digestive enzyme activity in the anterior intestine and oxidative stress markers in the liver. Fish in the AgNP 0.01 and 0.1 groups had the lowest anterior intestinal fold and enterocyte height. However, there were no statistically significant changes in the digestive enzyme activity in the anterior intestine. Analysis of enzymatic activity in the liver showed an increase in superoxide dismutase activity in fish in the AgNP 0.1 group. Histological analyses showed that AgNPs inhibited meiotic divisions at prophase I in a non-linear manner in ovaries and testes. In the AgNP 0.1 and 1.0 groups, the area occupied by spermatocytes was lower compared to the other groups. These results indicate that exposure to AgNPs may lead to disturbances in morphology and enzymatic activity in the liver and intestine and may lead to disruption of reproduction in populations.

## 1. Introduction

Silver nanoparticles (AgNPs) are released into the environment as a result of both natural and anthropogenic processes. The second is more significant—the effect of environmental contamination as a consequence of the use of nanomaterials. They pollute both water and terrestrial areas at every stage of the product life cycle [1,2,3,4,5,6,7,8]. To date, the toxic effects of silver nanoparticles and gold nanoparticles on gonads and fertility, among others, have been described [9,10,11,12], including the disruption of meiotic divisions during spermatogenesis and oogenesis, induction of oxidative stress, genotoxicity, endocrine disruption, and stimulation of apoptosis in ovarian follicle cells [12,13].

The effects of nanoxenobiotics on gonadal development and fertility of fish are still poorly understood, mainly because of the use of research fish models characterised by oviparity, which is the most basic reproductive strategy among this group of animals. Viviparity is a type of reproduction that evolved from oviparity to increase the survival rate of offspring [14] and involves embryonic development in the mother’s body. One type of viviparity is matrotrophic viviparity, wherein the embryo is nourished by the placenta or other structures that perform similar functions. Goodeidae is a family of viviparous Cyprinodontiformes fishes whose natural occurrence is limited to the Mexican Plateau. It is a small family that, depending on the systematic view, includes 38 [15] to 51 (fishbase.org, accessed on 28 August 2023) species of fish, most of which are endemic, of which butterfly splitfin (*Ameca splendens*) is used as a model fish in scientific studies [16,17,18]. The butterfly splitfin is characterised by its small body size, with adults reaching body lengths of approximately 75–100 mm [19]. They are omnivorous fish, although under natural conditions their main source of nutrients is algae or small invertebrate animals such as mosquito larvae, copepods, or oligochaeta [19]. This species is commonly maintained in aquaria as an ornamental fish (fishbase.org, accessed on 28 August 2023). In addition, the histological structure of the ovaries has been studied and revealed asynchronous development, indicating the occurrence of multiple reproductive cycles in this species. This allows for the obtaining of numerous offspring, usually between 11 and 17 individuals per litter, and sometimes even more, depending on storage conditions [19]. All viviparous species belonging to the family Goodeidae and, with the exception of *Ataeniobius toweri*, have a unique structure called the ‘trophotenic placenta’, a type of pseudoplacenta that has the function of absorbing nutrients from the mother [20,21].

Another area under study is the effect of nanoproducts on gastrointestinal physiology. Changes in liver morphology and physiology as a result of the negative effects of NPs on this organ have been previously described [22,23,24], but the effects of NPs on the gut remain poorly described. Dias de Cunha and Brito-Gitirana [25] observed that exposure of zebrafish (*Danio rerio)* to titanium dioxide nanoparticles can lead to abnormal morphology of intestinal mucosal cells, including vacuolization. In addition, such changes as shedding of cells and necrosis of mucosal cells as well as degeneration of mucosal cells.were observed in the intestines of common carp (*Cyprinus carpio*) exposed to AgNPs long term [26]. Moreover, Nile tilapia (*Oreochromis niloticus*) fed a diet enriched with sodium butyrate nanoparticles showed increased intestinal fold length/width, crypt depth, surface absorption area, and number of goblet cells compare with the control group [27].

As suggested by Wang et al. [28], nanoproducts can also affect the activities of various enzymes, including digestive enzymes. These authors showed that copper nanoparticles can inhibit protease, amylase, and lipase activity in the digestive tract. However, serum amylolytic, proteolytic, and lipolytic activities were shown to increase in Nile tilapia fed feed with sodium butyrate nanoparticles [27]. These contradictory results regarding the effect of nanoparticles on enzymatic activity demonstrate that their impact is not fully understood and requires further research.

The exemplary studies by other authors indicate that the toxic effect of NPs can be observed across the entire organism of the fish, leading to disruption of homeostasis and even death of the individual [29,30].

Therefore, the aim of this study was to determine the long-term effects of silver nanoparticles coated with polyvinylpyrrolidone (AgNPs PvP) on the morphology and enzymatic activity of a model fish species—butterfly splitfin (*Ameca splendens*). The effects of the nanoxenobiotic on the histological structure of the liver and anterior intestine were analysed, as well as the activity of digestive enzymes in the intestine and markers of oxidative stress in both organs. In addition, the effects of the AgNPs on spermatogenesis and oogenesis were histologically analysed.

## 2. Results

During the experiment, no mortality was observed in any of the experimental or control groups (Figure 1). On the last day of the experiment, there were no statistically significant differences in the body weight, body length, or Fulton’s fitness index (Figure 1).

### 2.1. Histology of Anterior Intestine

The analysis of histological structure of the anterior intestine revealed that the AgNP 1.0 group showed the longest anterior intestinal folds, the height of which differed significantly (*p*  ≤  0.05) compared to the values of this parameter in the other AgNP-exposed groups (Figure 2). The structure of the anterior intestinal wall in all groups was preserved with a properly formed mucosa containing numerous goblet cells. Enterocytes in the groups of fish exposed to AgNPs concentrations of 0.01 mg/dm^3^ and 0.1 mg/dm^3^ showed statistically significantly (*p*  ≤  0.05) lower heights of enterocytes compared to fish in the other experimental groups (Figure 2). Similarly, the lowest heights of supranuclear areas were observed in the AgNP 0.01 and 0.1 groups, and the values in these groups were significantly (*p*  ≤  0.05) different compared to those in the control and AgNP 1.0 groups (Figure 2). The highest supranuclear height (*p*  ≤  0.05) was found in the group of fish exposed to AgNP 1.0 mg/dm^3^ (Figure 2).

In the experimental groups, there was local detachment of the epithelium from the middle *lamina propria* at the top of the intestinal folds, along with the presence of lymphocytes or small inflammatory infiltrates in the mucosa, compared to the control group (mainly lymphocytes, eosinophils, and macrophages) (Figure 3A–D). Analysis of the width of the *lamina propria* in the anterior intestinal folds showed that fish exposed to AgNPs 0.01 and 0.1 mg/dm^3^ had the narrowest *lamina propria* compared to the other experimental groups, and the differences were statistically significant (*p*  ≤  0.05) (Figure 2). At the base of the intestinal folds there was infiltration of inflammatory cells, mainly lymphocytes, macrophages, and eosinophilic cells. Macrophages were also present in the mucosa and submucosa throughout their height, along with deposits of pigment granules in the cytoplasm (Figure 3B–D).

### 2.2. Histology of Liver

Histomorphometric measurements showed that the fish in the AgNP 1.0 mg/dm^3^ group had the highest average cross-sectional area through hepatocytes; however, there were no statistical differences in the values of this parameter between the experimental groups (Figure 3). The livers of the studied fish were characterized by varying *steatosis* of hepatocytes. The predominant type of *steatosis* was microvesicular *steatosis*, with some individuals in the group of fish exposed to AgNP 1.0 mg/dm^3^ also showing white adipose tissue in the exocrine pancreatic parenchyma adjacent to the liver (Figure 4A–D). The normal morphological and histological structure of the liver was preserved in the control group and the AgNP 0.1 mg/dm^3^ group. In the other groups, a disordered cytoarchitecture of the liver parenchyma, consisting of hepatocytes diffusion and compressed sinusoids, was observed. In all analysed fish from all experimental groups, small eosin and PAS-positive granules were present in the cytoplasm of hepatocytes, indicating the presence of glycogen (Figure 4). Yellow pigment granules and melanomacrophage centres were also visible among the hepatocytes, especially at higher concentrations of AgNPs (Figure 4).

### 2.3. Histology of the Ovaries

Morphological analysis revealed no pathological changes in the female gonads. Oogonia and oocytes at different stages of development were observed in the ovarian cortex (Figure 5). The total number of female germinal cells counted and summed in the 6 tested females from each experimental group differed between groups. The highest number of female germinal cells (489 cells) was found in the ovaries of fish in the control group. A lower number was found in the AgNP 0.01 and AgNP 1.0 groups (300 cells and 305 cells, respectively), while the lowest number of germinal cells was observed in females from the AgNP 0.1 group (128 cells).

Histomorphometric analysis showed no changes in the number of oogonia (0–25 µm diameter) between the control group and the AgNP 0.01 and AgNP 0.1 groups, where oogonia accounted for more than 10% of all germinal cells in the female gonads (Figure 5). The AgNP 1.0 group had fewer oogonia than the other groups and accounted for 7.2% of all germinal cells in the ovary (Figure 5). The highest proportion (above 70%) of previtellogenic oocytes was found in the gonads of fish from all experimental groups (Figure 5). The largest number of oocytes of this stage was observed in the gonads of fish exposed to AgNP 0.1. However, the distribution of stages 1, 2, and 3 was unequal (Figure 5). The largest number of oocytes at stage 2 (50–110 µm) and stage 3 (110–175 µm) was observed in the AgNP 0.1 group, while this experimental group had the fewest oocytes at stage 1 (Figure 5), where the proportion was more than twice as low compared to the other experimental groups. In contrast, the fewest oocytes at stage 3 were observed in sections throughout the ovaries of fish in the AgNP 0.01 group (Figure 5). Vitellogenic oocytes were the least frequently observed in the gonads of fish exposed to AgNP 0.1 mg/dm^3^, where their proportion was 5.4% (Figure 5). Very few stage-5 oocytes were found in these gonads and no stage-6 oocytes were observed (Figure 5).

### 2.4. Histology of the Testes

The presence of spermatogonia, spermatocytes, and spermatids was observed in the cross-sections of the testes of the analysed fish from all experimental groups. In addition, morphological analysis did not reveal any pathological changes in the structure of the testis or the ultrastructure of male germinal cells in fish from all investigated groups (Figure 6E–J). Ultrastructural analysis revealed spermatogonia located at the base of the seminal tubules, which were mitotically divided (Figure 6G,J). Spermatocytes were located inside the seminal tubules above the areas occupied by the spermatogonia (Figure 6F,I). No changes indicative of pathological meiosis were observed in any of the experimental groups. Furthermore, microscopic image analysis of spermatids and spermatozoa revealed no abnormalities in sperm head formation, chromatin condensation, or strand formation (Figure 6E,H). These germinal cells were located above the area occupied by the spermatocytes and were closest to the seminal duct.

However, AgNPs affected the population of male germinal cells. The area occupied by spermatogonia was the highest in the testes of fish from the AgNP 1.0 group (Figure 7). The largest area occupied by spermatocytes in the testis was observed in sections through the male gonad of the butterfly splitfin exposed to AgNPs at a concentration of 0.01 mg/dm^3^ (more than 60% of the area), while the smallest was in the AgNP 1.0 group (less than 40% of the area). The largest area occupied by spermatids was found in the gonads of fish in the AgNP 0.1 group (54.4% of the area), whereas the smallest area was found in the AgNP 0.01 group (33.7% of the area) (Figure 7).

### 2.5. Enzymatic Activity in Anterior Intestine

Analysis of digestive enzyme activity showed no statistically significant differences between the experimental groups (Figure 8). Regarding the ACP, higher activity was found in the intestines of fish in the AgNP 0.01 and AgNP 1.0 groups (Figure 8). In contrast, ALP activity decreased with increasing concentrations of the xenobiotic to which the fish were exposed (Figure 8). Amylolytic (amylase), lipolytic (lipase), and proteolytic (trypsin) activities did not differ between the experimental groups. There was only a slight reduction in digestive activity in fish in the AgNP 0.1 group, but no statistically significant differences were found (Figure 8). In addition, a large variation in the activities of these digestive enzymes was observed between the analysed individuals in the experimental group (Figure 8).

### 2.6. Enzymatic Activity in Liver

In the liver parenchyma of fish exposed to the tested AgNPs, an increase in ACP and ALP activity was observed depending on the concentration of the tested xenobiotic. In the case of ACP, statistically significant differences (*p*  ≤  0.05) were found in the activity of this enzyme between the AgNP 0.01 and AgNP 0.1 groups (Figure 9). On the other hand, ALP expressed the highest statistically significant activity (*p*  ≤  0.05) of this enzyme in the livers of fish from the AgNP 1.0 group compared to the control group (Figure 9). AgNP concentration did not affect GPX activity, whereas there was a statistically significant increase (*p*  ≤  0.05) in SOD activity in the AgNP 0.1 group compared to fish in the control and AgNP 0.01 groups (Figure 9).

## 3. Discussion

The rapid development of nanotechnology applies mainly to the food, textile, and construction industries, as well as to medicine, cosmetology, pharmaceuticals, and others. A particularly popular nanomaterial is AgNPs, which are characterised by antiseptic and antibacterial properties, among others [31]. In recent years, the global production of nanosilver has exceeded 550 tons per year [32]. This increase in production, and the popularity of AgNPs in various spheres of life, raises the possibility of interaction of this nanoxenobiotic with the terrestrial and aquatic environments, which may also affect human health. As suggested by Ferdous and Nemmar [33], the effects of AgNPs on humans as well as the environment are not fully understood; therefore, it is necessary to explore the existing gap related to the risk assessment of exposure to AgNPs. It is particularly important to know the effects of long-term exposure to AgNPs at the molecular, organism, and ecosystem levels [34,35]. In particular, industrial wastewater removal and erosion of engineering materials in household products are examples of the potential for their introduction into the environment [36]. It is estimated that up to 800 t of AgNPs may be released into water bodies worldwide each year [37]. This results in predicted environmental concentrations of AgNPs of approximately 10–1800 ng/L in surface water and 40–80,000 µg/kg in sediments [32]. Therefore, it is important to investigate the effects of AgNPs on butterfly splitfin. The obtained results indicate that exposure to AgNPs may lead to disturbances in morphology and enzymatic activity in the liver and anterior intestine and may lead to disruption of meiotic divisions in the testes and ovaries. It was observed that higher concentrations of AgNPs could interfere with the meiosis, as previously observed by the authors in earlier studies [12]. However, the effects of AgNPs on gastrointestinal histology and digestive enzyme activities are still poorly understood, and the results presented here provide a better understanding of this phenomenon.

The mortality of organisms is a basic parameter that indicates the toxic effects of the xenobiotics. In the present study, no mortality was observed among butterfly splitfins subjected to long-term exposure to AgNPs. Moreover, no statistically significant differences were observed in the other growth parameters of the fish: weight, total and standard length, or condition factor. The results obtained may be surprising, since many publications indicate that metal nanoparticles can benefit the growth of various fish species [38,39,40] or are toxic to aquatic organisms. This negative effect can manifest as mortality, with a linear relationship not always found between increasing AgNPs concentrations and increased mortality [12] or inhibition of fish growth [32,41].

Many factors influence fish growth, but food intake and digestive processes in the digestive tract play a crucial role. To date, the effects of AgNPs on the digestive physiology of various fish species are poorly understood. Analysis of the morphology of the anterior intestine of the butterfly splitfin showed statistically significant lower intestinal folds in groups exposed to AgNPs at concentrations of 0.01 and 0.1 mg/dm^3^, as well as lower enterocytes and narrower *lamina propria* compared to the values of these parameters in fish from the control group. This affects the reduction of the absorption surface area and, consequently, can lead to malnutrition. Similarly, Shahare et al. [42] found that AgNPs affected the mucosa of the small intestine of mice by damaging the brush border and intestinal glands. These histopathological changes can significantly reduce the absorption of nutrients from the intestinal lumen. In contrast, the values of these parameters were statistically significantly higher in the AgNP 1.0 experimental group. This lack of linear toxicity of AgNPs has already been found in previous studies and may be related to the aggregation of nanoxenobiotics [12]. In addition, the use of PvP-coated AgNPs in the experiment affected their stabilization and limited their potential to bind to proteins, making this nanoxenobiotic less available to the organism (bioavailability) [43].

Similarly, a reduction in the efficiency of absorption of food digestion products during exposure of common carp to AgNPs was observed by Kakakhel et al. [26]. In addition, Khorshidi et al. [44] observed that AgNPs reduce lipase and ALP activity and, consequently, lower growth rates. In contrast, no statistically significant differences were observed in the activity of the enzymes studied in the tested butterfly splitfins. However, in general, the activity of digestive enzymes is subject to strong variation between individuals, which may suggest the plasticity and adaptability of the enzyme apparatus to changing conditions, not only nutritional, but also environmental. Moreover, as Mwaanga et al. [45] suggest, the effect of sublethal doses of nanoxenobiotics may affect biochemical parameters faster than morphology, physiology, and growth parameters.

Unlike in the anterior intestine, the effects of AgNPs on liver function are better understood. The following were most commonly observed in the liver parenchyma of fish exposed to silver and other metal nanoparticles: changes in the size of hepatocytes—often associated with their vacuolization—necrosis, formation of melanomacrophage centres, and infiltration of immune cells or increased deposition of Browicz–Kupffer cells [22,24,46,47]. In the tested fish, changes in the surface area of liver cells were observed (the smallest in the control group, the largest in the AgNP 1.0 group, with no statistically significant differences), vacuolisation of hepatocytes was observed, and melanomacrophage centres were formed in the AgNP-exposed groups. This indicates a toxic effect of the studied nanoxenobiotic on liver homoeostasis, although some of the histopathological changes were not correlated with the AgNP concentration. On the other hand, an increase in the activity (without statistically significant differences found) of ACP and ALP in the livers may indicate damage to this organ [48]. Many authors have reported changes in the activity of enzymes responsible for antioxidant protection during exposure to nanoparticles [12]. In the fish studied, a statistically significant increase in SOD activity was found in the liver in the AgNP 0.1 mg/dm^3^ group. It is the enzyme responsible for deactivating free radicals formed in the body [49], becoming the first line of defence against oxidative stress. In contrast, in the group exposed to the lowest concentration of AgNPs, comparable to that modelled in the environment, SOD activity was comparable to that found in fish from the control group. The increase in SOD activity in the livers of fish exposed to high concentrations of AgNPs compared to the level of activity found in the control group is in accordance with the observations of Vrček et al. [50]. As found in human hepatoma cells in vitro, the cells exposed to silver ions and AgNPs are characterised by depletion of GSH activity, increased production of ROS, and increased SOD activity. This may indicate progressive oxidative stress. However, in the viviparous species *Chapalichthys pardalis,* which is related to the butterfly splitfin, Valerio-García et al. [51] observed a decrease in SOD activity, as well as GPX activity. In the analysed butterfly splitfins, SOD activity increased, while GPX activity was not statistically significantly different, although lower activity than that found in the control group was observed in fish exposed to AgNPs at concentrations of 0.01 and 0.1 mg/dm^3^. These observations may confirm the nonlinear activity of the studied PvP-coated AgNPs and their tendency to agglomerate, in accordance with the mechanism proposed by Szudrowicz et al. [12].

The effects of nanoparticles on the oogenesis in various fish species are still poorly understood. It has been shown that exposure of fish to aqueous solutions of metal nanoparticles can disrupt meiotic divisions [12] but also induce cytotoxicity and genotoxicity [10] in the female gonad. In the butterfly splitfins studied, it was shown that AgNPs at a concentration of 0.1 mg/dm^3^ had a strong effect on the process of oogenesis. In fish from this group, the highest percentage of previtellogenic oocytes was observed compared to the other experimental groups. However, there was a clear imbalance between the proportion of stage 1 and stage 2 oocytes, indicating the inhibition of meiotic divisions in the first stages. In contrast, the proportion of oocytes at advanced stages of meiotic division (stages 5 and 6) was the lowest compared to that in the other experimental groups, indicating an obvious inhibition of oogenesis and a reduction in reproductive potential. This strong effect of AgNPs at a concentration of 0.1 mg/dm^3^ was also expressed in a drastic reduction in the total number of female germinal cells in the gonads of fish from this experimental group. Similar observations were described by Orbea et al. [52], who found that the exposure of zebrafish to PVP/PEI-coated AgNPs caused a decrease in the number of eggs laid by females. It is possible that the long-term exposure of the tested butterfly splitfins to silver nanoparticle solutions may have affected the hormonal regulation of oogenesis. As indicated by Degger et al. [9], AgNPs affect the process of steroidogenesis in fish, which, as a consequence, may lead to the disruption of the secretion of hormones responsible for the regulation of oogenesis. As suggested by Szudrowicz et al. [12], this may affect fertility disorders in the fish populations.

Many authors have indicated that AgNPs can penetrate the male gonad and can infiltrate the blood/nucleus barrier, affecting spermatogenesis and reproductive potential [53,54]. Histological analysis showed no pathological changes in the male gonad and did not confirm the presence of AgNPs in the testis. Similar observations regarding the structure of the seminal tubules in the male gonads of the zebrafish were shown by Szudrowicz et al. [12]. Similarly to oogenesis, during the development of male germinal cells in the testes of fish exposed to silver nanoparticle solutions, many authors observed the formation of oxidative stress, the development of pathological changes, and the disruption of sex hormone secretion [55,56,57]. In addition, disorders of spermatogenesis, spermiogenesis, and oocyte fertilization are often observed [12,55]. In the current study, AgNPs were shown to affect spermatogenesis in the butterfly splitfin. The decrease in the proportion of spermatids in the male gonads of fish exposed to the two highest concentrations of AgNPs (0.1 and 1.0 mg/dm^3^) is particularly pronounced. This may be due to the depletion of the meiotic potential of germinal cells during prolonged exposure, as confirmed by Thakur et al. [55] in the testes of rats exposed to AgNPs. In contrast to the observations of Fathi et al. [56] in the gonads of male rats, no decrease in the number of spermatogonia was observed in the animals exposed to AgNPs. In the fish studied, there was even an increase in the proportion of spermatogonia in the testes of butter splitfin exposed to AgNPs 1.0 mg/dm^3^, compared to fish in the control group and others exposed to AgNPs. This effect of AgNPs in the tested fish may be related to the tendency of the nanoparticles used in the experiment to agglomerate, which reduces the toxicity of the nanoproduct [12].

## 4. Materials and Methods

### 4.1. Scheme of the Experiment

The experiment was performed with the approval of the 2nd Local Ethical Committee for Animal Experiments at the Warsaw University of Life Sciences (approval number WAW2/009/2019, dated 30 January 2019).

The study material consisted of adult, sexually mature butterfly splitfin (*Ameca splendens*) of approximately 5 months of age. The fish were kept in 15-L tanks at a density of 2.5 individuals/dm^3^. Prior to the experiment, the fish in the aquaria underwent a seven-day acclimatization. After acclimatisation, the fish were divided into four groups according to the concentration of polyvinylpyrrolidone (PvP)-coated silver nanoparticles (AgNPs PvP) to which they were exposed—0.01 mg/dm^3^, 0.1 mg/dm^3^, and 1.0 mg/dm^3^ for 42 days. Individuals in the control group were maintained in water without tested xenobiotics. Each experimental group was conducted in 3 replicates with 6 individuals in each tank—4 females and 2 males (n = 18 in each experimental group). Half of the water volume in the tanks was changed every 24 h, and the level of xenobiotics was supplemented to the initial concentration. During the experiment, fish were fed twice a day ad libitum with commercial feed: TetraMin (Tetra GmbH, Herrenteich, Germany) or Spirulina Forte (Tropical, Chorzów, Poland) every other day at 9 a.m. and frozen Artemia at 4 p.m. During the experiment, in accordance with the breeding requirements of the species, the temperature difference between day and night was used (the average daily temperature was 23.0 ± 3.2 °C). The total hardness of the water was in the range of 6–10 °n, while the carbon hardness reached 5–6 °n. During the experiment, the pH of the water was 6.8 ± 0.2. No nitrite (NO^2−^) was detected in the water, and the amount of nitrate (NO^3−^) was less than 20 mg/dm^3^. The day/night light regime lasted for 12 h each. At the end of the experiment, the survival rates were assessed, and then all fish from each group were anaesthetized with MS-222 solution (tricaine methanesulfonate, 3-amino-benzoic acid ethyl ester, Sigma-Aldrich, St. Louis, MO, USA) and decapitated. The material was collected for histological (light and electron microscopy) and biochemical analyses. The fish were measured and weighed before decapitation, and, based on these measurements, the Fulton condition factor was calculated using the formula [58]
(1)$$K=\frac{m}{Si^{3}}\times 100$$
where:

*m*—body weight (g),*Si*—body length (mm).

### 4.2. Characterization of Nanoparticles

The polyvinylpyrrolidone (PvP)-coated silver nanoparticles (AgNPs PvP) used in the study were purchased from Sigma Aldrich (Schnelldorf, Germany, part number 576832-5G; less than 100 nm in diameter). This nanoxenobiotic has been used in previous studies. Nanoparticle dilutions were prepared according to the previously described method: AgNPs were suspended in water at 1000 mg/dm^3^ concentration and sonicated for 3 cycles of 15 min each using Ultron U-505 ultrasonic cleaner (Ultron, Dywity, Poland) at 45 °C. To confirm the characterisation of nanoparticles in solution, NPs and their agglomerates were analysed by transmission electron microscopy, DLS analysis, and zeta potential analysis according to the methods described by Szudrowicz et al. (2022) [12]. The results of these analyses are presented in Table 1. The results of the analyses confirmed that the studied AgNPs’ PvPs had a diameter smaller than 100 nm, and the diameters of their agglomerates were larger than 200 nm (Table 1). The zeta potential ranged from −25 to −20 mV (Table 1). A detailed description of the analysis of AgNPs has been presented by Szudrowicz et al. (2022) [12].

### 4.3. Histological Analyses

For histological analyses, six specimens of whole female ovaries and male testes and nine digestive systems were taken from each study group. The material was fixed in Bouin’s solution and subjected to a standard histological procedure: dehydration in an increasing range of ethanol concentrations. The material was cleared in xylene and embedded in paraffin. The samples were sectioned into 5 μm thick slides (Leica RM 2265 microtome, Leica Microsystems, Wetzlar, Germany). The histology of livers, intestines, and gonads was evaluated based on slides stained with hematoxylin and eosin (H&E). The glycogen was detected using the combined method of alcian blue (pH 2.5) and periodic acid and Schiff (AB/PAS).

Histomorphometric analysis of the sections through the H&E-stained male gonads included measurements of the areas occupied by spermatogonia, spermatocytes, and spermatids in 15 separate seminal tubules of 6 males from each experimental group. Meanwhile, in the ovaries of all females (*n* = 6) subjected to the histological analysis, the diameter distribution of oogonia and oocytes in the ovaries was measured. The results of the female germinal cell diameter measurements allowed us to distinguish the classes describing the maturity of germinal cells. The division into classes for the development of oocytes of the butterfly splitfin was adopted based on the scale developed by Tinguely (2015) [15] for the related species *Xenotoca eiseni*, as shown in Table 2. In addition, the total number of germinal cells visible on cross sections through the ovaries was counted in all females studied. Moreover, an analysis of the digestive tract was performed. On H&E-stained liver sections: the area of the hepatocytes (µm^2^) (100 measurements in each individual); H&E-stained intestinal sections: the height of the intestinal folds (10 measurements in each individual), the height of the enterocytes (50 measurements in each individual), and the height of the supranuclear surface of the enterocytes (50 measurements in each individual).

All measurements were performed using a Nikon Eclipse Ni-E and Nikon Eclipse 90i microscope (Nikon, Tokyo, Japan) and the image analysis program QuPath (v0.3.0) [59].

The analyses of testis ultrastructure were performed on fragments that were fixed in 2% (*w*/*v*) of paraformaldehyde and 2.5% (*v*/*v*) of glutaraldehyde solution in a 0.05 M cacodylate buffer (pH 7.2) for 2 h. Sections were prepared, embedded in resin, and cut on an ultramicrotome according to the methodology described by Szudrowicz et al. (2022). Ultra-thin slices on meshes were examined on an FEI 268D “Morgagni” transmission electron microscope (FEI Company, Hillsboro, OR, USA) equipped with an Olympus-SIS “Morgagni” digital camera (Olympus, Münster, Germany).

### 4.4. Detection of Enzymatic Activity

In order to determine the effect of AgNPs on enzymatic activity in the livers and anterior intestines of the tested fish, enzymatic analyses were performed. Specimens from 10 individuals from each group were collected (*n* = 10); livers and anterior intestines sections were frozen in liquid nitrogen and stored at −80 °C. The collected tissues were then homogenised in deionized water at 4 °C and centrifuged for 10 min at 14,000× *g* at the same temperature. The supernatant was collected, and then the tissues were refrozen in liquid nitrogen and stored at −80 °C. In livers, the activities of alkaline phosphatase (ALP), acid phosphatase (ACP), superoxide dismutase (SOD), and glutathione peroxidase (GPX) were examined [60,61]. In the anterior intestine, the activities of alkaline phosphatase (ALP), acid phosphatase (ACP), amylase, trypsin, and lipase were measured according to the methodologies described by Palińska-Zarska et al. [60] and Wiszniewski et al. [61]. The activity results were divided by the protein concentration (measured using the biuret method) in the sample and expressed as U/g. The determinations were conducted in 96-well plates, and enzymatic activity was determined from absorbance readings at 37 °C using a Tecan microplate spectrophotometer (Infinite 200 PRO; Tecan, Männedorf, Switzerland); measurements were made in triplicates.

### 4.5. Statistical Analysis

Quantitative results from measurements from all organs and body indices were subjected to a test for consistency with a normal distribution (Shaphiro-Wilk test). To test for statistically significant differences between groups, data were subjected to one-way ANOVA with post hoc Tuckey test or, for non-parametric data, Kruskall–Wallis test. All tests were performed in STATISTICA software (version 13.0).

## 5. Conclusions

Based on the results obtained, it can be concluded that 42-day exposure of butterfly splitfin to AgNPs induces morphological changes in the anterior intestine and liver, disrupts spermatogenesis and oogenesis, and generates oxidative stress. The strongest pathological changes were not correlated with the highest-tested concentration of the nanoxenobiotic, which confirms previous observations regarding the nonlinear toxicity of AgNPs. The analysis of the results shows that even the effect of low concentrations of AgNPs, referring to those modelled in the environment, affects fish homeostasis and reproductive potential. The described results of the analysis are the first report indicating the effect of low concentrations of AgNPs during prolonged exposure on a model species that is intriguing from the perspective of toxicological studies—the butterfly splitfin.

## Figures and Tables

**Figure 1 ijms-24-14598-f001:**
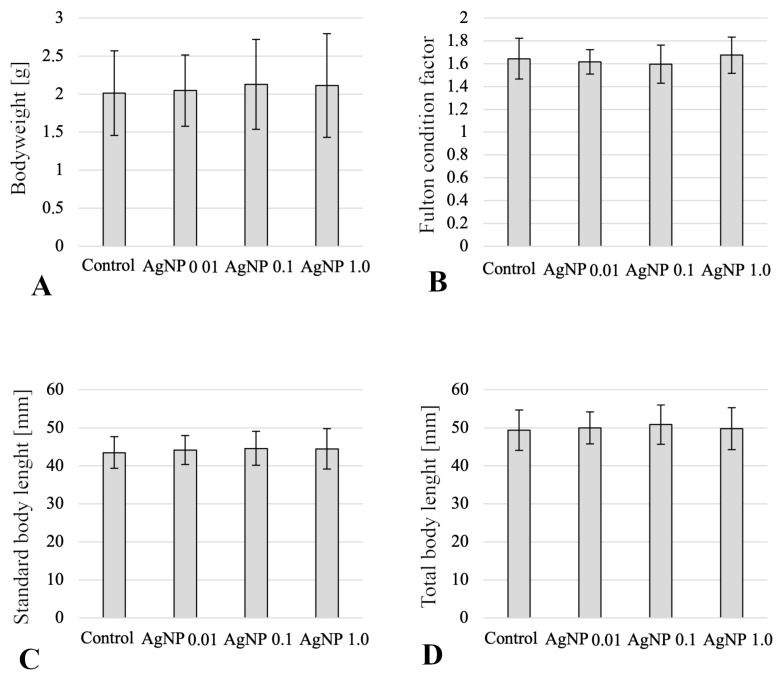
Morphometric measurements of butterfly splitfin treated with AgNPs from 0.01 to 1.0 mg/dm^3^: (**A**) body weight (g); (**B**) Fulton condition factor; (**C**) standard body length (mm); (**D**) total body length at the end of the experiment (mm).

**Figure 2 ijms-24-14598-f002:**
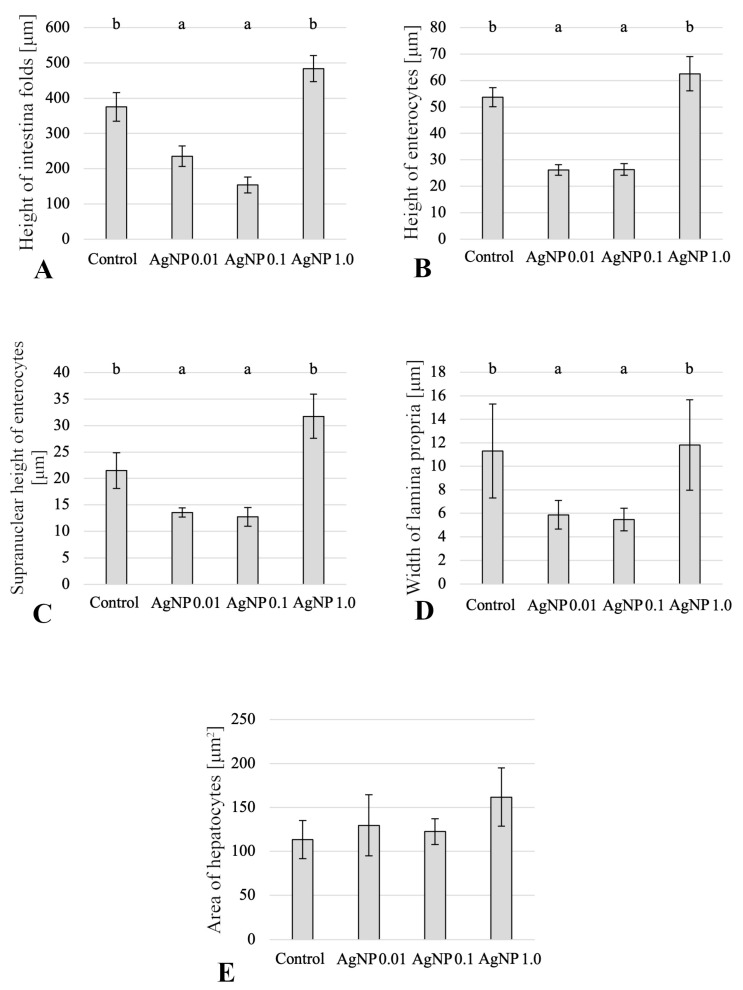
The histomorphometric parameters of fish anterior intestines and livers on the last day of the experiment: (**A**) height of anterior intestinal folds (μm); (**B**) height of enterocytes (μm); (**C**) supranuclear height of enterocytes (μm); (**D**) width of *lamina propria* (μm); (**E**) area of hepatocytes (μm^2^). Mean values ± SD are shown. Different letters indicate the statistical differences between groups (*p*  ≤  0.05).

**Figure 3 ijms-24-14598-f003:**
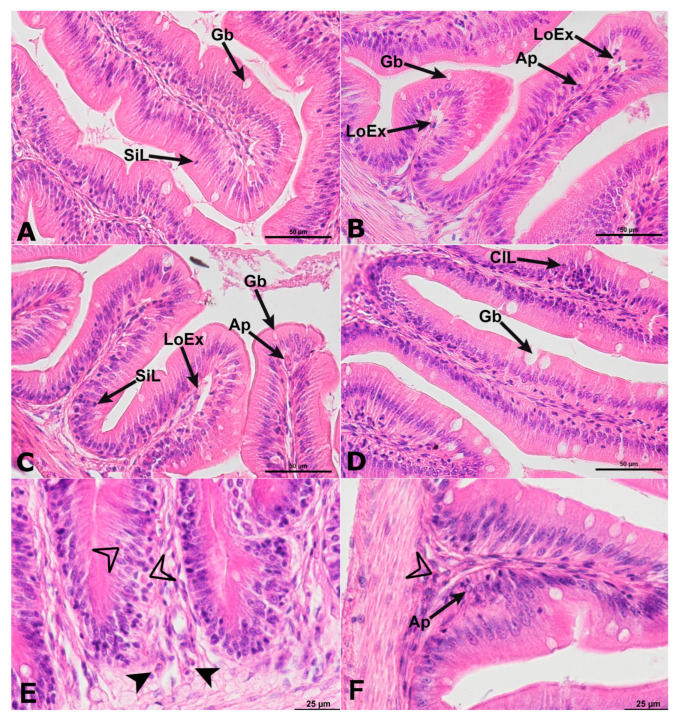
The histological structure of the anterior intestine in butterfly splitfin from the control group (**A**) and the groups treated with AgNPs from 0.01 mg/dm^3^ (**B**); 0.1 mg/dm^3^ (**C**); and 1.0 mg/dm^3^ (**D**). Lesions of varying severity were observed in all fish in the experimental groups. In control fish, goblet cells (Gb) and singular lymphocyte infiltration (SiL) were observed in the intestinal mucosa. The mucosa of the experimental fish was often infiltrated with lymphocytes (singular—SiL, or as a cluster—ClL), apoptotic-like cells in the apical of folds (Ap), local exfoliation (LoEx). Details of histopathological changes and immune cells in the anterior intestine of fish from the AgNP 0.1 (**E**) and AgNP 1.0 (**F**) group. Eosinophilic immune cells (filled arrowhead) and macrophages (empty arrowhead). H&E staining, scale bars—50 µm (**A**–**D**); scale bars—25 µm (**E**,**F**).

**Figure 4 ijms-24-14598-f004:**
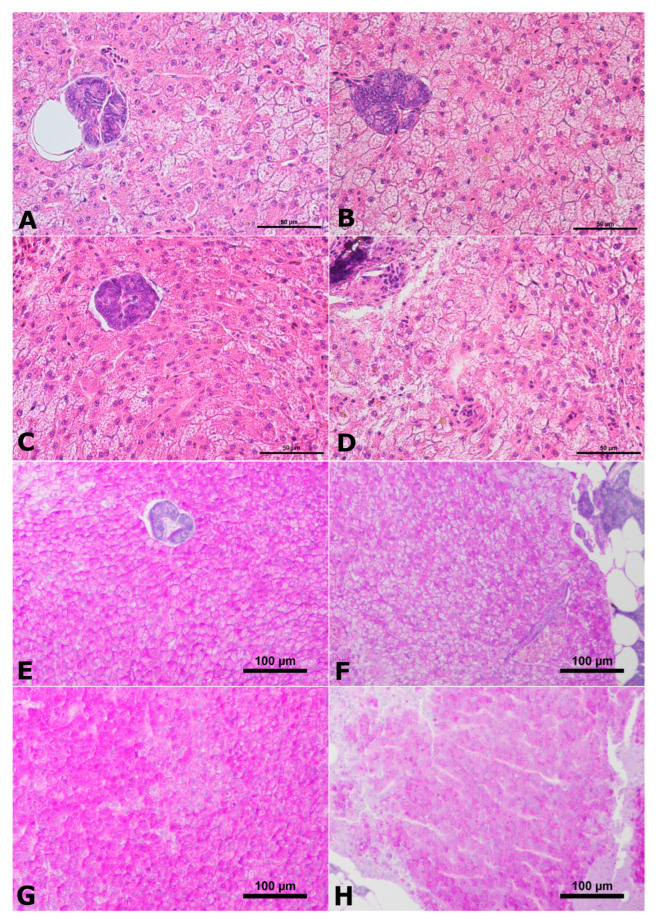
Cross-section through the liver of the butterfly splitfin from (**A**) control group; (**B**) AgNP 0.01 group; (**C**) AgNP 0.1 group; (**D**) AgNP 1.0 group. (**E**) Control group; (**F**) AgNP 0.01 group; (**G**) AgNP 0.1 group; (**H**) AgNP 1.0 group. H&E staining (**A**–**D**); AB/PAS staining (**E**–**H**). Scale bars 50 µm (**A**–**D**); scale bars 100 µm (**E**–**H**).

**Figure 5 ijms-24-14598-f005:**
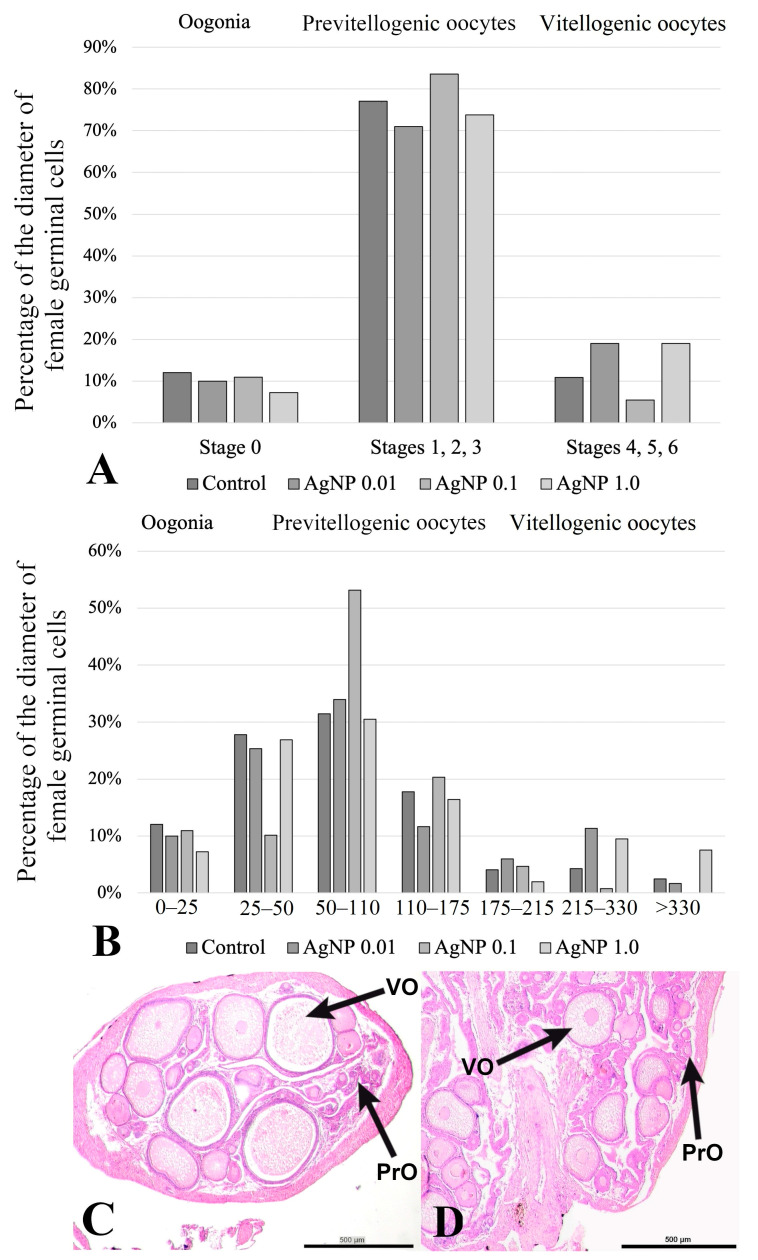
Morphology and histomorphometry of the female gonad butterfly splitfin. (**A**) Percentage of the diameter of female germinal cells (oogonia, previtellogenic oocytes and vitellogenic oocytes) in the experimental groups studied; (**B**) percentage of the diameter of female germinal cells according to developmental stages described by Tinguely [15]. Sections through the female gonad: (**C**) from the control group; (**D**) from the AgNP 1.0 group. PrO—previtellogenic oocytes; VO—vitellogenic oocytes. H&E staining. 500 µm scale bar.

**Figure 6 ijms-24-14598-f006:**
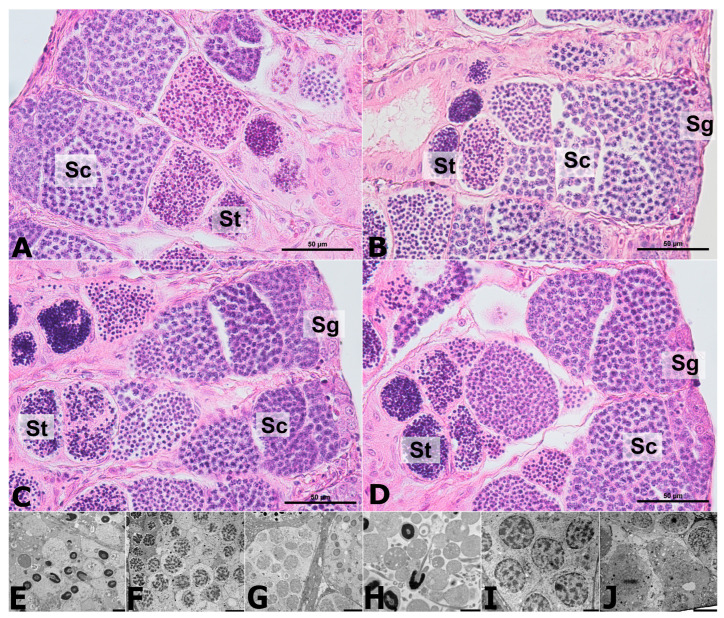
Cross-section through the male gonad of the butterfly splitfin from: (**A**) Control group; (**B**) AgNP 0.01 group; (**C**) AgNP 0.1 group; (**D**) AgNP 1.0 group. Ultrastructure of male germinal cells of fish from the control group (**E**–**G**) and AgNP 1.0 group (**H**–**J**)—spermatids (**E**,**H**), spermatocytes (**F**,**I**), and spermatogonia (**G**,**J**). Sc—spermatocytes; St—spermatids; Sg—spermatogonia. H&E staining (**A**–**D**); TEM scans (**E**–**J**). 100 µm scale bar (**A**–**D**); 5 µm scale bar (**E**–**J**).

**Figure 7 ijms-24-14598-f007:**
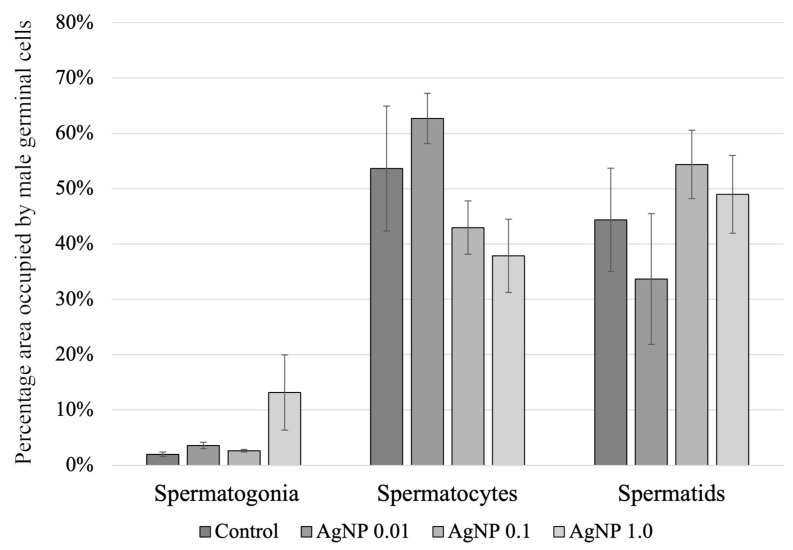
Percentage of testicular area occupied by different groups of germinal cells.

**Figure 8 ijms-24-14598-f008:**
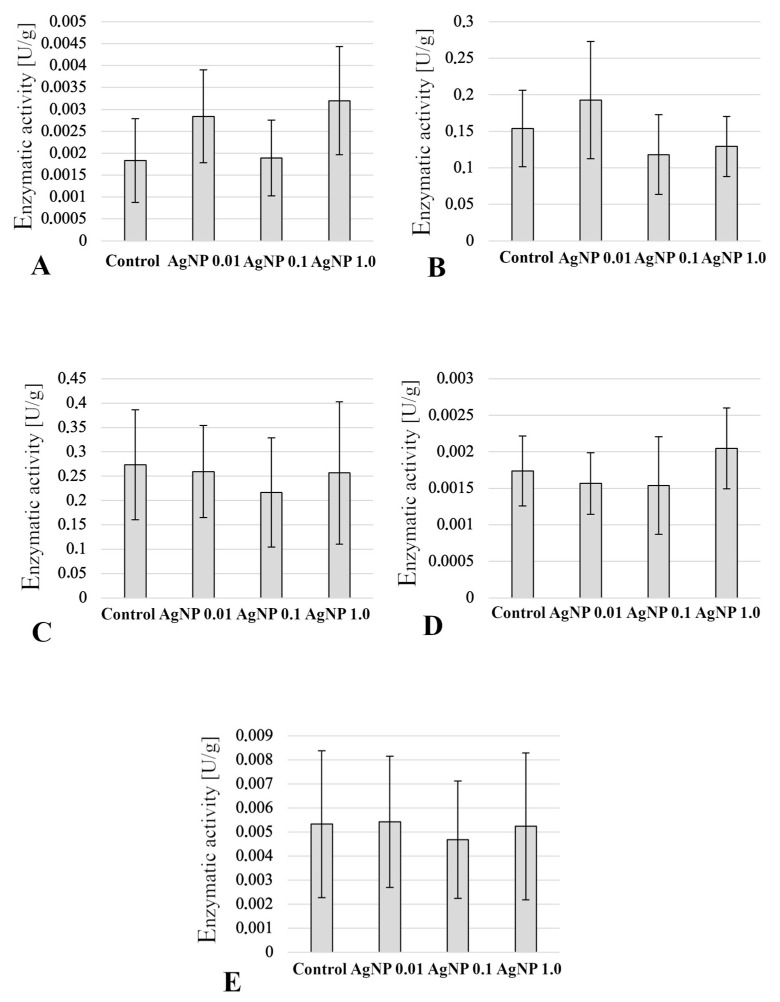
Digestive enzyme activity (U/g) in the anterior intestine (mean values ± SD): (**A**) ACP—acid phosphatase; (**B**) ALP—alkaline phosphatase; (**C**) Amylase; (**D**) Lipase; (**E**) Trypsin.

**Figure 9 ijms-24-14598-f009:**
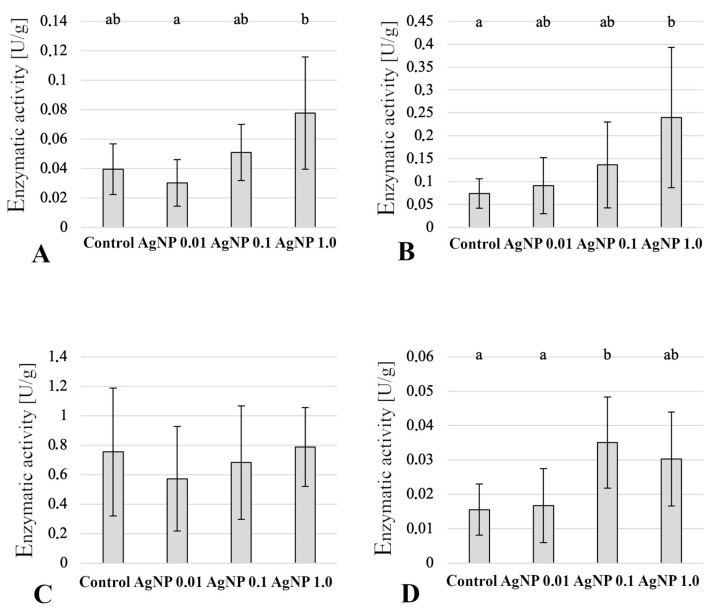
Enzymatic activity in the liver (mean values ± SD): (**A**) ACP—acid phosphatase; (**B**) ALP—alkaline phosphatase; (**C**) GPX—glutathione peroxidase; (**D**) SOD—superoxide dismutase. Different letters indicate statistical differences between groups (*p*  ≤  0.05).

**Table 1 ijms-24-14598-t001:** Characterisation of the AgNPs used in the experiment, according to Szudrowicz et al. (2022) [12].

Parameter	Value
Hydrodynamic diameter range (DLS analysis)	min. 39 nm; max. 93 nm
Diameter of agglomerates (DLS analysis)	>202 nm
Diameter range (TEM analysis)	min. 6.14 nm; max. 108.79 nm
Diameter of agglomerates (TEM analysis)	>200 nm
Mean of zeta potential (in 25 °C)	−23.4 mV

**Table 2 ijms-24-14598-t002:** Developmental stages of female germinal cells of butterfly splitfin based on Tinguely (2015) [15].

Stage		Name	Diameter (µm)
Oogonia	0	Oogonia	0–25
Primary growth (previtellogenic)	1	Early primary growth	25–50
	2	Mid primary growth	50–110
	3	Late primary growth	110–175
Secondary growth (vitellogenic)	4	Early secondary growth	175–215
	5	Late secondary growth	215–330
	6	Fully growth oocytes	above 330

## Data Availability

Not applicable.

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
