# Peer review of "The Effect of Silver Nanoparticles on the Digestive System, Gonad Morphology, and Physiology of Butterfly Splitfin (Ameca splendens)"

_ijms, 2023, doi:10.3390/ijms241914598_

Round 1

Reviewer 1 Report

Comments and Suggestions for Authors

Dear Authors,

the experimental part was carried out in a very detailed manner in the paper but the results need to be improved. The weakest part of the work is the Results. I suggest you add physicochemical characterization of the AgNPs. Did you use positive control AgNO3? If yes please the results to the figures. However, there are some mistakes in the writing of the paper itself, and all suggestions for changes were entered directly into the manuscript. 

Author Response

Dear Reviewer 1,

Thank you for your review of our publication "The effect of silver nanoparticles on the digestive system and gonads morphology and physiology of butterfly splitfin (Ameca splendens)". We appreciate your valuable input and interest in this work. As suggested by the reviewer, punctuation and grammatical changes have been made in the main text, which were suggested in the PDF file sent with the review. Particular attention was paid to the insertion of ‘.’ instead of "," when writing decimal fractions. Below, we provide detailed responses to each comment.

  1. Please add the physico-chemical characterisation of AgNP

As suggested by the reviewer, the detailed results of the nanoparticle analyses performed have been added to Table 2.

  1. nor in

Changed from „and” to „nor in”.

  1. use one term through the text.

The term “AgNPs” was used throughout the text.

  1. Affected?

Changed from “damaged” to “affected”.

  1. in this study

This phrase has been added to the revised sentence.

  1. what about positive control? AgNO3?

Previous tests have been carried out in which butterfly splitfins were exposed to silver ions derived from AgNO3. The ions were found to be highly toxic to this species, with the consequence that not many fish were able to survive a week's exposure. Therefore, following the recommendations of Poland Local Ethical Committee II in Warsaw, the main experiment was conducted without using a group of fish exposed to AgNO3.

  1. number?

The reference number has been added.

Reviewer 2 Report

Comments and Suggestions for Authors

The authors have used the enzymatic activity and histopathology to determine the fish health after exposed the nanoparticle. Overall form this manuscript, some/major errors and unsuitable data are strongly noted in the results. Some lesions such as the presence of eosinophilic immune cells, apoptotic cell and macrophages were observed by H and E staining method or showed as a low magnification. They are not accepted. Advance method such as TUNEL assay should be supported to exist the apoptotic cell in the tissue. No labels are commonly found in several figures. These situations are strongly impacted in the result part under the error data. Therefore, in conclusion, the paper is not suitable for publication and the authors are encouraged to collaborate with a pathologist with training in this area of study.  Also, a great deal of the text/results simply re-states the presence of normal features - and I am unsure of how this would add to the scientific literature as this material is easily found elsewhere.

Comments on the Quality of English Language

Some errors for your English have been found.

Author Response

Dear Reviewer 2,

Thank you for reviewing our publication titled "Effect of silver nanoparticles on the morphology and physiology of the digestive tract and gonads of the butterfly splinfin (Ameca splendens).” We appreciate the valuable input and interest in our work. Numerous comments have improved the manuscript to improve its readability. As the reviewer pointed out, it would be beneficial if the results were consulted with a pathologist to confirm the observed changes. We were able to request the expertise of a histopathologist specializing in ichthyopathology, affiliated with the European Association of Fish Pathologists (EAFP), who helped us edit the description of the results.

Below are detailed responses to each comment.

  1. The title was changed from „The effect of silver nanoparticles on butterfly splitfin (Ameca splendens) digestive system and gonads morphology and physiology” to „ The effect of silver nanoparticles on the digestive system, gonad morphology and physiology of butterfly splitfin (Ameca splendens).” according to reviewer suggestions.
  2. pls specific data what is the physiology ? (enzymatic activity ? histopathological response?)

As suggested by the reviewer, the term 'physiology', which was used in the text, was clarified and referred to as 'enzymatic activity.’

  1. pls add what are the selected organs that the author used ? pls specific organs/tissues?

Selected organs used in the analysis were added.

  1. Suggested" The enzymatic activity should be normal more than the enzyme activity.

The term has been changed in accordance with the reviewer’s suggestion.

  1. pls more explanation for this phase!

The inhibition of meiotic division occurs during prophase I in both male and female gonads.

  1. what the area of spermatocytes ? the authors mean "size cell?

The parameter measured in the male gonad was the area occupied by spermatogonia, spermatocytes, and spermatids but not the cell area. The term "area" was clarified in the abstract.

  1. The enzymatic data and nanoparticles ? they are missing !

A section has been added to the 'Introduction, ’ in which the differential effects of nanoparticles on enzymatic activity are briefly described.

  1. References are required.

References were added.

  1. Changed from „fish fertility” to „fertility of fish”

Change has been made.

  1. Some and short data about this fish biology are required. they help to support for reader!

A short description of the Goodeidae family and Ameca splendens has been added to the Introduction section.

  1. more histopathological alterations are required.

A paragraph describing the histological changes associated with exposure to nanoparticles has been added.

  1. Specific references are need.

References were added.

  1. specific! what is the physiology ?

As suggested by the reviewer, the term 'physiology', which was used in the text, was clarified and referred to as 'enzymatic activity.’

  1. Using histological method?

Added “histologically analysed”

  1. what the authors are proposed for this condition! and references are required.

The purpose of the study was clarified; therefore, following the reviewer's suggestion, the section indicating the effect of nanoparticles on population parameters that were not analyzed was removed.

  1. First step !! the author should be explained the intestinal regions and their composition! what are different among them?

The description of the anterior intestinal sections used for histological analysis has been rewritten. In the 'results' section, there is a reference to the fact that the histological results relate to the anterior intestine analysis.

  1. what the author is referred "no significant histopathology changes? you explained "normal intestinal histology?

This sentence has been corrected accordingly.

  1. (pls add the valve ?)

P-value has been added to the statistically significant differences

  1. the colour narrows are not Ok they are not easy to follow anf I think that the authors should be showed as a abbreviations.

The labels in the figures have been changed to make them more legible.

  1. pls used "the experimental fish" Is OK

Changed „experimental animals” to „experimental fish” as suggested by reviewer

  1. The apoptotic cell should be proved by TUNEL assay. Only H and E staining method is not accepted !.

Of course, the reviewer is correct for the diagnosis of apoptotic cells, and H&E staining is not sufficient. Admittedly, it allows the observation of so-called apoptotic bodies, structures typical of programmed cell death, but such observations should be supported by other stains. The authors did not have the possibility to perform TUNEL analysis, as suggested by the reviewer, but they attempted to diagnose apoptotic cells using an immunohistochemical technique to detect the active chain of caspase 3, which is a marker of cell apoptosis (one of the effector caspases). Staining was performed with the necessary controls but did not show a specific antibody-binding site in the intestines of the butterfly splendens. In this situation, the authors changed the description of the pathological changes in the anterior intestine using the following phrase: "apoptotic-like cells". In the Discussion, suggestions were added that the effect of AgNPs on the anterior intestinal mucosa should be verified in the future.

Below are examples of two photos of the anterior intestine of the test fish, one after IHC reaction with antibody and the other after IHC reaction without caspase 3 antibody (negative control).

IHC staining with caspase 3 antibody

IHC staining without caspase 3 antibody (negative control)

  1. I am not believed these cell types based on the H and E staining method.

I saw to indicate that Eosinophilic immune cells. It is very hardly indentified and I warranted that this is a  oval lymphocytes ! whereas the macrophage !! what is the cretirial data that the author focused!. TEM and IHC should be focused and supported for this data.

We acknowledge the difficulty in distinguishing eosinophilic immune cells from oval lymphocytes using this method alone, which might result in a false diagnosis. H&E staining can be used to identify eosinophilic particles in the immune cells. Therefore, observations of the appearance of different types of immune cells have been generalized and described as immune cells. In addition, figure 3 has been modified by adding histological images taken under higher magnification.

  1. pls add (Fig ?)

Added “Fig.4”.

  1. what is the small eosin?

This sentence refers to 'small eosin granules,’ which are mentioned in the next part of the sentence.

  1. PAS ? what is the figure ? and the author showed be showed the structure and composition as abbreviations and label for each figure/ Why the author did not label?

AB/PAS staining images of the liver have been added to Figure 4.

  1. ovary ?

Changed from “gonads” to “female gonads” to avoid repetition.

  1. what is the normal structure ?

This sentence has been revised accordingly.

  1. what is the germinal cell?

The sentence regarding reproductive cells has been clarified.

  1. 489?

“489 cells” has been added.

  1. what the small number?

The term “small numbe”' refers to the stated number of cells in the following sentence, and was changed to “lower” and “lowest number”.

  1. The explanation for Y axis should be corrected in all figures.

A description of the Y-axis has been added to the figures.

  1. pls correct these step as figure ? previtellogenis stage???? or previtellogenic stgae ??

The correct terms are "previtellogenic" and "vitellogenic" oocytes. The term in the figure has been changed accordingly.

  1. labels were required to correlate with the resulting data.

In figure 5, the following symbols have been added.

  1. More explanation from TEM study should be noted

The histological description of the male gonad, including TEM, was extended, and a reference was made to the images presented in figure 6. The following paragraph was added: “Ultrastructural analysis revealed spermatogonia located at the base of the seminal tubules, which divided mitotically (Fig. 6 G, J). Spermatocytes were located inside the seminal tubules above the areas occupied by the spermatogonia (Fig. 6 F, I). No changes indicative of pathological meiosis were observed in any of the experimental groups. Furthermore, microscopic image analysis of spermatids and spermatozoa revealed no abnormalities in sperm head, chromatin condensation, or strand formation (Fig. 6 E, H). These germinal cells were located above the area occupied by the spermatocytes and were closest to the seminal duct.”

  1. what?

We changed “tested xenobiotic” to “AgNPs.”

  1. Labels for each figure should be showed, which they are correlated the content/results.

The correct labels were added.

  1. what your results are proposed?

The first paragraph has been enriched with a general reference to the results. A detailed discussion of the results follows.

  1. Reference is required.

Reference has been added.

  1. Previtellogenic, Vitellogenic stage?

Changed from “previtellogenesis” and “vitellogenesis” to “prewitellogenic” and “vitellogenic”.

  1. Observation of enzymatic activity?

Changed from “Biochemical analysis” to “Detection of enzymatic activity”

As suggested by the reviewer, punctuation and grammatical changes have been made in the main text, which were suggested in the PDF file sent with the review. Particular attention was paid to the insertion of ‘.’ instead of "," when writing decimal fractions.

Round 2

Reviewer 1 Report

Comments and Suggestions for Authors

Dear authors, the manuscript was improved according to all recommendations, therefore I suggest accepting it.

Kind regards.

Author Response

Dear Reviewer 1,

Thank you for your review of our revised publication " The effect of silver nanoparticles on the digestive system, gonad morphology and physiology of butterfly splitfin (Ameca splendens)". Thank you for reviewing the manuscript and accepting the changes.

Best regards,

Maciej Kamaszewski

Reviewer 2 Report

Comments and Suggestions for Authors

Dear,

       I happy to see the revised manuscript that it is nice when compared to previous observation. However, only small sub-figures in Figure 4 should be showed as high magnification and rewrite A-F!. All figures should be explained.

Best regards,  

Comments on the Quality of English Language

All regions are clear, but the re-checked ms should be corrected again.

Author Response

Dear Reviewer 2,

Thank you for reviewing our revised publication titled "The effect of silver nanoparticles on the digestive system, gonad morphology and physiology of butterfly splitfin (Ameca splendens).” As suggested, figure 4 has been restructured, with magnified AB/PAS images placed under the letters E-H. The caption under the figure has also been changed and new references to this figure have been added in text.

Best regards,

Maciej Kamaszewski